# Interaction between Single Nucleotide Polymorphisms (SNP) of Tumor Necrosis Factor-Alpha (TNF-α) Gene and Plasma Arsenic and the Effect on Estimated Glomerular Filtration Rate (eGFR)

**DOI:** 10.3390/ijerph19074404

**Published:** 2022-04-06

**Authors:** Yi-Jen Fang, Kuan-Lin Lin, Jyuhn-Hsiarn Lee, Kuei-Hau Luo, Tzu-Hua Chen, Chen-Cheng Yang, Hung-Yi Chuang

**Affiliations:** 1Ph.D. Program in Environmental and Occupational Medicine, College of Medicine, Kaohsiung Medical University, Kaohsiung 807, Taiwan; u101803001@kmu.edu.tw (Y.-J.F.); lukaslee@nhri.edu.tw (J.-H.L.); 2Digestive Disease Center, Show-Chwan Memorial Hospital, Changhua 500, Taiwan; 3Department of Public Health, College of Health Sciences, Kaohsiung Medical University, Kaohsiung 80708, Taiwan; u106575106@kmu.edu.tw; 4National Institute of Environmental Health Sciences, National Health Research Institutes, Miaoli County 350, Taiwan; 5Graduate Institute of Medicine, College of Medicine, Kaohsiung Medical University, Kaohsiung 807, Taiwan; u107800007@kmu.edu.tw; 6Department of Family Medicine, Kaohsiung Medical University Hospital, Kaohsiung 807, Taiwan; 1030578@kmuh.org.tw; 7Department of Occupational Medicine, Kaohsiung Municipal Siaogang Hospital, Kaohsiung Medical University, Kaohsiung 812, Taiwan; u106800001@kmu.edu.tw; 8Department of Occupational and Environmental Medicine, Kaohsiung Medical University Hospital, Kaohsiung Medical University, Kaohsiung 807, Taiwan; 9Research Center for Environmental Medicine, Department of Public Health and Environmental Medicine, College of Medicine, Kaohsiung Medical University, Kaohsiung 807, Taiwan

**Keywords:** arsenic, estimated glomerular filtration rate, TNF-α gene polymorphisms

## Abstract

When poisons enter the human body, tumor necrosis factor (TNF-α) will increase and cause damage to tissues through oxidative stress or inflammatory reaction. In previous studies, arsenic (As) has known to cause many health problems. Some studies have shown that As exposure is negatively correlated with estimated glomerular filtration rate (eGFR), or with the prevalence of proteinuria. At present, there are few studies focusing on the effects of As exposure and TNF-α single nucleotide polymorphism (SNP) to eGFR; thus, this study was intended to explore the interactions between TNF-α SNPs and plasma As and their effects on eGFR. A cohort of 500 adults, aged 30 to 70 years, was randomly selected from Taiwan Biobank (TWB). We used the gene chip to screen out seven SNPs of the TNF-α gene and used the results, combined with questionnaires, biochemical tests, and stored plasma samples from the TWB, for the analysis of As by inductively coupled plasma mass spectrometry (ICP-MS). After adjustments for BMI, hypertension, hyperlipidemia, kidney stones, and smoking habits, multiple regression statistics were performed to explore the interaction between SNPs and plasma As with eGFR. In this sample of the general population, plasma As had a significant association with the decline of eGFR (β (SE) = −7.92 (1.70), *p* < 0.0001). TNF-α gene SNP rs1800629 had the property of regulating TNF-α, which interacts with plasma As; individuals with the AG type had a significantly lower eGFR than those with the GG type, by 9.59 mL/min/1.73 m^2^ (*p* < 0.05), which, regarding the dominant model, could infer that the A allele is a risk allele. SNP rs769177 had no interaction with plasma As; however, participants with the TT or TC type had significantly higher eGFR levels than the CC carriers, by 4.02 mL/min/1.73 m^2^ (*p* < 0.05). While rs769176 interacted with plasma As, if a person with the TC type had a higher plasma As concentration, that would sustain higher eGFR. This study found that certain SNPs of the TNF-α gene would be robust to the decline of eGFR caused by As exposure. Still, we need further research to confirm the protective regulation mechanism of these SNPs.

## 1. Introduction

Previous studies have shown that arsenic (As) exposure can cause kidney damage as low as 0.2 mg/kg/day of inorganic As (lowest observed adverse effect level, LOAEL) [1], but the detailed mechanism of the suffering was still unclear. In recent years, studies have shown that As induced apoptosis through oxidative stress response and cytokine regulation, thereby damaging the distal convoluted tubules of the kidney [2,3]. Some studies have found that the metabolites of As, monomethyl arsenic acid (MMA) and dimethyl arsenic acid (DMA), have a greater impact on kidney function than inorganic As [4,5]. Most of the previous studies focused on the analysis of As in urine, and less on the blood As. Whether urinary or blood As, many studies have shown that As exposure is inversely related to the glomerular filtration rate (GFR) or to the prevalence of proteinuria; however, the results were not consistent [5,6,7]. Furthermore, studies have revealed that chronic As exposure would lead to positive regulation of tumor necrosis factor- alpha (TNF-α) mRNA and other related factors, and then to atherosclerosis and skin lesions. [2,8]. The mechanism was believed to be related to the destruction of endothelial cells; TNF-α molecules also regulate the inflammatory reactions through this pathway [2,8,9].

The gene that regulates TNF-α is located on chromosome 6 (chromosome 6p21.3) and is part of the human major histocompatibility complex (MHC class III) segment [10]. Although many studies have shown that TNF-α is related to the decline of GFR or renal function, the specific mechanism is still unclear [5,11,12]. Some research findings have revealed that the increase in TNF-α induces additional reactive oxygen species (ROS) production, which can lead to inflammatory reactions and oxidative stress and damage vascular endothelial cells [13,14]. Another study found that TNF-α could promote renal vasoconstriction, resulting in the reduction of GFR, and directly inhibit the ability of distal convoluted renal tubular epithelium to transport sodium ions [15].

Studies have found that whether it is As in blood or urine, there is a positive correlation with the level of TNF-α. Th issue is not only about total As; more in-depth study has examined the relationship between methylation products of As metabolism and TNF-α [16]. Based on the role of TNF-α in the pathophysiology, it is reasonable to infer that any substance that is toxic to the body will induce TNF-α. Therefore, some researchers have sought to find genotypes that may regulate TNF-α production, and then predict the impact of exposure to such toxic substances in people with different genotypes. The currently known TNF-α production regulatory site is located in the promoter of the gene, and there is an SNP rs1800629 (or TNF-308 G/A) [17,18,19]; in addition, a study on the relationship between As exposure and TNF-α gene polymorphism also reflected that the rs1800629 would play a very important role.

Based on previous studies, we hypothesized that arsenic exposure would induce an increase in TNF-α to disrupt glomerular function, resulting in a decrease in eGFR. In view of the fact that few studies have mentioned the effect of arsenic exposure and TNF-α SNP types on GFR in humans, the goal of this study was to further explore the relationship among As, TNF-α SNP types, and the influence of GFR (the index of kidney function).

## 2. Materials and Methods

This study was a cross-sectional study of data obtained from the Taiwan Biobank (TWB) of Academia Sinica, on a selection of members from the general population representing normal controls in other studies in Taiwan (56% males and 44% females, aged 30 to 70 years; for details, please refer to https://www.twbiobank.org.tw (accessed on 28 March 2022). We applied to use the data and stored plasma samples of 500 people that were randomly selected by frequency matching according to age and gender of the general population in Taiwan, aged 30 to 70 years. In addition, the 500 subjects in this study were from the general population, excluding individuals with cancers, auto-immune diseases, and other catastrophic illnesses. The study was approved by the Institutional Review Board of Kaohsiung Medical University Hospital (KMU-HIRB-E(I)-20150259, initial date of approval: 6 January 2016) and approval was waived for individual consent forms, due to de-identification in Taiwan BioBank data and specimens.

Blood and biochemical exam findings from the Taiwan Biobank were all analyzed at the Linkou Chang Gung Memorial Hospital; in addition, each subject was administered a questionnaire on their demographic information, personal health behavior, and present medications. Plasma As in stored plasma from these same 500 subjects of TWB was analyzed by the laboratory in the Research Center for Environmental Medicine, Kaohsiung Medical University, using inductively coupled plasma mass spectrometry (ICP-MS, Thermo Scientific XSERIES 2^®^, Waltham, MA, USA). The resolution of ICP-MS was set to 0.8 and 0.4 amu at 10% peak height, which complied with the typical As analysis. Radio Corporation of America (RCA) cleaning standard was used on all equipment in the laboratory. We used standard reference materials (SRMs) to conduct repeated analyses for quality assurance (QA) and quality control (QC). Each result had to fit the reference between 90% and 110%. QC was to ensure the stability of the system by triple testing the SRM sample, of which the coefficient of variance (CV) should be less than 3%.

For data on genotypes in the TWB, we used the Axiom Genome-Wide Array Plate chip system (called Taiwan Biobank chip (https://taiwanview.twbiobank.org.tw/search) accessed on 28 March 2022) designed by Taiwan National Center for Genome Medicine (NCGM) in cooperation with Affymetrix, USA, to select a total of 653,291 SNPs, from which we searched for all SNPs of the TNF-α gene. Seven SNPs were found: rs1799964 and rs1800629, with function as 2 KB upstream variant; rs1800610 and rs3093662, with function as intron variant; rs3093668, with function as 500 B downstream variant; and rs769176 and rs769177, with an unknown function.

From the 500 selected individuals, we excluded one due to missing biochemical data and six others with heterozygosity rates that were over three standard deviations, which meant their DNA samples were contaminated or inbred. Finally, 493 people were used for statistical analysis.

Many methods are used to estimate GFR. In recent years, many European and American studies have tended to recommend the CKD EPI formula, which is accurate when estimating for relatively healthy people with high glomerular filtration rates and relatively small deviations [20]; additionally, it can also reduce the overdiagnosis of CKD [21].

For representation of continuous variables, we used the mean ± standard deviation (SD) and quartiles (median and interquartile range, IQR). Categorical variables were represented by the number (percentage). Plasma As measurements were log-transformed to generate normal distribution. The generalized linear regression model [22] was used to explore the relationship between the plasma As and the TNF-α gene polymorphisms and their effect on eGFR while controlling for confounding factors. Finally, the interaction term of plasma As and SNPs were added to the generalized linear regression models to explore the effects of interaction between plasma As and SNP on eGFR after controlling confounding factors. Every regression model contained each SNP; thus, there were models 1 to 9. The interaction term was significant, and a plot of interaction was generated. All statistical analyses were carried out with SAS 9.4 (SAS Institute Inc. Cary, NC, USA). A two-tailed *p*-value < 0.05 was considered significant.

## 3. Results

There were 275 males (55.78%) and 218 females (44.33%) in the study; the age distribution ranged from 30 to 70 years, with an average of 48.27 years. Table 1 lists the factors such as smoking habits and chronic diseases that may affect eGFR, based on the TWB questionnaire. There were 132 participants (26.77%) with smoking habits; 28 with diabetes (5.68%); 69 with hyperlipidemia (14.00%); 56 with hypertension (11.36%); and 36 with kidney stones (7.30%). The mean and SD of plasma As were 4.04 and 1.79 (µg/L), respectively. Since the distribution of plasma As was skewed, the natural logarithm (ln) was used to correct the skewness and also used in the regression analyses. The last part of Table 1 lists the CKDEPI eGFR calculated according to the formula. The average eGFR was 101.36; all individuals in this study population fell within the normal range of the first stage CKD (eGFR ≥ 90 mL/min/1.73 m^2^).

There were seven SNPs of the TNF gene on the TWB chip. Table 2 showed the eGFR and plasma As levels of the study population according to each SNP type. Of the seven SNPs, rs1799964 and rs1800629 were located in the gene regulatory region. SNPs rs1800629 and rs3093668 did not meet Hardy-Weinberg equilibrium; however, rs3093668 had only one homozygote variant (CC type). Similarly, rs769177 contained only one homozygote variant (TT type). For the last four SNPs in Table 2, due to the small number of homozygote variant types, the homozygote and heterozygote variant types were combined for calculation in the regression analyses.

Table 3 shows the regression model of the effect of natural log-transformed plasma As (ln(plasma-As)) on CKDEPI eGFR without considering the influence of SNP, and adjusted for BMI, diabetes, hypertension, hyperlipidemia, kidney stones, and smoking habits. In addition to plasma As, BMI and hypertension had significantly inverse effects on eGFR (*p* < 0.05). The plasma As level, similarly to the previous studies, had a damaging effect on eGFR; for example, assuming that a person’s ln(plasma As) was reduced from Q3 to Q1 (lnQ3 − lnQ1 = IRQ = 0.45, e^0.45^ = 1.57 μg/L, actual plasma As reduced), then the CKDEPI eGFR could be increased by 1.57 × 7.92 = 12.43 (mL/min/1.73 m^2^).

Multiple regression analyses were performed to explore the relationship of each SNP and plasma As with CKDEPI eGFR, and the confounding factors such as BMI, diabetes, hypertension, hyperlipidemia, kidney stones and smoking habits were adjusted; then, the results were integrated into Table 4 (models 1 to 7). All ln(plasma As) showed a significant decrease in eGFR, with regression coefficients ranging from −7.74 to −8.20, which meant higher plasma As levels would correspond to lower CKDEPI eGFR. The regression coefficients of ln(plasma As) with the addition of individual SNPs were not significantly different from the regression coefficient (−7.92) of ln(plasma As), as shown in Table 3. Of the seven SNPs, only rs769177 was significantly associated with CKDEPI eGFR. Because only one participant had the TT type of rs769177, we combined (TT + TC) versus CC type and obtained the regression coefficient as β = 4.02 (*p* < 0.05), as shown in Table 4 for model 6.

All seven SNPs in this study were analyzed using interaction models that showed significant interactions only by rs1800629 and rs769176 with ln(plasma As) (Table 4, models 8 and 9). In the interaction models, no matter which SNP was involved in the action, the natural log-transformed plasma As showed a significant reduction effect on eGFR. Plasma As interacted with SNPs, which were more clearly presented in the figure.

Figure 1 shows the interaction between natural log-transformed plasma As and SNP: s(A) Interactions with rs1800629 types AA and GG were not significant; however, rs1800629 AG significantly interacted with plasma As levels to decrease CKDEPI eGFR. (B) There were only CC and TC types for rs769176; the cross lines show a significant interaction between these types and plasma As levels. Participants with the TC type of rs76179 would be more robust to As exposure than CC carriers; however, the numbers were too small.

## 4. Discussion

The study found that in the model without genotypes, the eGFR decreased by 7.92 mL/min/1.73 m^2^ (*p* < 0.0001) for each unit of increase in natural log-transformed plasma As. In addition, when SNPs (with or without interaction) were added into the regression models, each unit increase in ln(plasma As) was associated with a decrease in eGFR ranging from 6.47 to 8.29 mL/min/1.73 m^2^, all of which were significant. The results, similar to those of previous studies, supported the view that arsenic exposure harms kidney function even in individuals from the general population.

Our study used plasma As as a biomarker, in contrast to many other studies that used urine As for the exposure index [1]. While plasma As may have a shorter half-life, it could be an internal dose of exposure, whereas urine As could be based on both exposure and metabolic index. We used direct plasma As concentrations as internal doses of exposure in the cross-sectional study on the assumption they may be more appropriate for the exploration of interactions. Another reason was that the research participants were from the general population, not an occupationally exposed group. Direct plasma As concentrations may be sustained and consistent with previous environmental exposure; thus, changes in plasma As may be minor.

One of the main findings of this study was that the A allele of rs1800629(G > A) was susceptible to decreased eGFR. Since this SNP is located in the promoter of the TNF-α gene, it has attracted the attention of many researchers. As shown in some studies related to kidney function and As, carriers with the A allele of this SNP may show different expressions of the TNF gene, resulting in more susceptibility to skin lesions, eye and respiratory diseases induced by chronic arsenic exposure [23]; higher multiple organ failure (MOF score) assessments in patients with acute kidney injury and acute pancreatitis [19]; and higher contrast-induced nephropathy [OR = 2.11 (1.08–4.09), *p* = 0.025]. Creatinine clearance was also more significantly decreased compared to that in individuals with the GG type (0.88 ± 1.83 vs. 0.36 ± 0.70, *p* = 0.03) [24]. A study on diabetic nephropathy also revealed that patients with type AA had higher blood TNF-α concentrations than those with type GG, and A allele carriers were 2.1 times more likely to suffer from macroalbuminuria (OR: 2.1, *p* < 0.001) [17]. Our previous study of metal workers with occupational exposure also found that the concentrations of TNF-α in the blood of workers with AA/AG type were higher than that in the blood of GG type carriers [14].

The functions of rs769177 and rs769176 were unknown. This research found that people with rs769177 TT or TC types were robust against As-induced decreases in eGFR. Studies of these SNPs are rare. A Norwegian study found that rs769177 was significantly associated with breast density in postmenopausal women who had never received hormone therapy (*p* = 0.02) [25]. To our best knowledge and based on a search through PubMed.org (accessed on 28 March 2022), there is no related research on rs769176.

In this study, a total of seven SNP loci of TNF-α were obtained from the TWB v1.0 Chip (Taipei, Taiwan), among which rs1800629 and rs3093668 showed no Hardy–Weinberg equilibrium. Based on the Genome Reference Consortium (GRC) published 37th edition of the Human Genome (GRC37, reference version of TWB v1.0), the Hardy–Weinberg disequilibrium of rs3093668 was consistent with that of the Beijing Han ethnic group. Furthermore, all seven obtained SNP loci were analyzed in this study. If the α-value of 0.05 was corrected to 0.05/7 = 0.007, according to the Bonferroni correction, then rs3093668 could be in Hardy–Weinberg equilibrium (*p* = 0.0233).

The situation with rs1800629 was more sophisticated. There were two cases of Hardy–Weinberg disequilibrium: one was in the study of Caucasians with coronary artery disease (CABGGENOMICS: CABG_NORTHAMERICAN); the other was found in the data of Northern and Western Europeans in the HapMap project (Coriell Cell Repository (CCR) and the NIH Polymorphism Discovery Resource (NIHPDR)). According to the calculation by the Hardy–Weinberg formula, there were too many people with AA type in this study population. Additionally, our results showed that the A allele may be a risky allele that should gradually decrease in number through evolution; however, the average plasma As concentration of this type (AA) was the lowest (AA 3.66; AG 3.97; GG 4.07μg/L), which may explain why the AA type could be retained.

There were some limitations in this study. Many previous studies on arsenic exposure were conducted in high-exposure areas such as Bangladesh, where the average values of blood or urinary As were much larger than those in this study. Moreover, previous studies were less focused on blood arsenic, and our study focused on the plasma As of people in the general population; thus, it was a pity that there were few comparable data. Furthermore, since the cohort in this study comprised individuals from the general population who voluntarily participated, there may be a voluntary bias [26]. However, the outcome of this study concerned CKDEPI eGFR, which is an indicator of kidney function calculated from various biochemical values. Generally, as the participants did not know which values would be used to calculate eGFR, this limitation could not skew the results of our research. In addition, regarding the recall bias that may arise from the investigation of questionnaires including smoking habits, we compared the smoking rate of our sample to that in Taiwan, and found them to be similar (26.8% among our study subjects versus 21.8% in Taiwan) [27]. Furthermore, although we screened the SNPs at the genomic level and found that SNPs rs1800629 and rs769176 interacted with plasma As levels to influence kidney function, we did not study the regulatory mechanisms. Some researchers have revealed that chronic As exposure can lead to positive regulation of TNF-α mRNA and other related factors [2,4,8]. The implication is that there might be regulatory mechanisms in this case at the transcriptome level. More regulatory mechanism studies are needed in the future.

## 5. Conclusions

The present study revealed an association between decreasing kidney function and As concentrations in blood plasma, which was similar to findings from prior research. Furthermore, we found seven SNPs of TNF-α genes that might be involved in the regulatory mechanisms; however, this calls for more research at the transcriptome level in the future. Of the seven SNPs, rs1800629 and rs769176 interacted with plasma As levels to influence kidney function in the statistic regression models. The implication is that there might be regulatory mechanisms in this case at the transcriptome level. We suggest more mechanism-related studies of these two SNPs be performed in the future.

## Figures and Tables

**Figure 1 ijerph-19-04404-f001:**
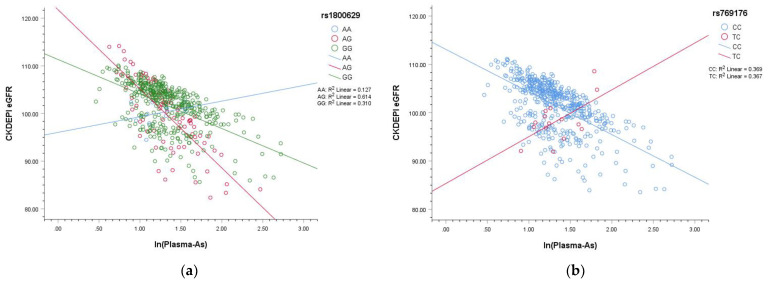
Plots showing interaction of natural log-transformed plasma arsenic (As) and SNPs. (**a**) Interactions with rs1800629 types AA and GG were not significant; however, rs1800629 AG significantly interacted with plasma As levels to decrease CKDEPI eGFR. (**b**) There were only CC and TC types for rs769176; the cross lines show a significant interaction between these types and plasma As levels. Participants with the TC type of rs76179 would be more robust to As exposure than CC carriers; however, the numbers were too small, and there were no participants with the TT type of rs769176.

**Table 1 ijerph-19-04404-t001:** Characteristics of study population.

Factors	Mean (±SD)/*n* (%)	Median	IQR
Sex			
Male	275 (55.78%)	-	-
Female	218 (44.33%)	-	-
Age	48.27 (±10.91)	47	14
BMI	24.43 (±3.46)	24.15	4.21
Blood pressure			
Systolic BP	115.24 (±17.51)	112	22
Diastolic BP	71.73 (±11.31)	70	16
Smoking			
Non-smoking	361 (73.23%)	-	-
Smoking	132 (26.77%)	-	-
Chronic disease			
Diabetes	28 (5.68%)	-	-
Hyperlipidemia	69 (14.00%)	-	-
Hypertension	56 (11.36%)	-	-
Kidney stones	36 (7.30%)	-	-
Blood test			
BUN ^a^ (mg/dL)	13.05 (±3.48)	12.6	4.5
Creatinine (mg/dL)	0.78 (±0.19)	0.79	0.29
HbA1c ^a^ (%)	5.66 (±0.55)	5.6	0.4
Plasma-As ^a^ (µg/L)	4.04 (±1.79)	3.59	1.69
ln(Plasma-As)	1.33 (±0.36)	1.28	0.45
Kidney function			
CKDEPI eGFR	101.36 (±14.25)	102.74	18.06

^a^: BUN (Blood Urea Nitrogen), HbA1c (Glycated Hemoglobin), As (Arsenic).

**Table 2 ijerph-19-04404-t002:** CKDEPI eGFR and plasma As levels of the study population according to each SNP type.

SNP	Number (%)	H–W ^a^	MAF ^a^	CKDEPI eGFR ^b^	Plasma As ^b^
rs1799964 ^I^	493	0.1498	0.18		
C/C	20 (4.06%)			101.28 (±13.59)	3.89 (±2.09)
C/T	134 (27.18%)			102.03 (±13.90)	4.40 (±2.12)
T/T	339 (68.76%)			101.11 (±14.45)	3.91 (±1.61)
rs1800629 ^I^	493	0.0023	0.11		
A/A	13 (2.64%)			100.21 (±9.99)	3.66 (±0.88)
A/G	85 (17.24%)			99.98 (±14.72)	3.97 (±1.59)
G/G	395 (80.12%)			101.70 (±14.27)	4.07 (±1.85)
rs1800610 ^II^	492	0.8703	0.17		
A/A	13 (2.64%)			104.10 (±16.59)	4.31 (±2.12)
A/G	137 (27.79%)			99.39 (±15.50)	4.15 (±1.97)
G/G	342 (69.37%)			102.03 (±13.59)	3.99 (±1.71)
rs3093662 ^II^	493	0.2132	0.03		
G/G	1 (0.20%)			120.21 (-)	2.34 (-)
G/A	23 (4.67%)			100.76 (±16.61)	4.07 (±1.09)
A/A	469 (95.13%)			101.35 (±14.13)	4.05 (±1.82)
rs3093668 ^III^	493	0.0233	0.02		
C/C	1 (0.20%)	0.7141 ^c^		120.21 (-)	2.34 (-)
C/G	15 (3.04%)			97.09 (±16.31)	3.99 (±1.07)
G/G	477 (96.75%)			101.46 (±14.16)	4.05 (±1.81)
rs769177 ^IV^	493	0.4242	0.06		
T/T	1 (0.20%)			110.84 (-)	4.06 (-)
T/C	62 (12.58%)			104.90 (±13.45)	3.85 (±1.71)
C/C	430 (87.22%)			100.83 (±14.31)	4.07 (±1.81)
rs769176 ^IV^	491	0.7311	0.02		
T/T	0 (0%)			-	-
T/C	15 (3.04%)			98.33 (±13.83)	3.95 (±1.12)
C/C	476 (96.55%)			101.40 (±14.26)	4.05 (±1.81)

Function: ^I^: 2 KB Upstream Variant; ^II^: Intron Variant; ^III^: 500 B Downstream Variant; ^IV^: None. ^a^: H–W (Hardy–Weinberg Equilibrium); MAF (Minor Allele Frequency), ^b^: CKDEPI eGFR (mL/min/1.73 m^2^); Plasma As (μg/L), ^c^: Combined (CC + CG) types versus GG type.

**Table 3 ijerph-19-04404-t003:** Multiple regression of plasma As on CKDEPI eGFR without genotypes.

CKDEPI eGFR	β (SE)
ln(Plasma-As)	−7.92 (1.70) **
BMI	−0.41 (0.18) *
Diabetes	
Diabetes (vs. Non-Diabetes)	−5.00 (2.66)
Hypertension	
Hypertension (vs. Non-Hypertension)	−7.08 (1.56) **
Hyperlipidemia	
Hyperlipidemia (vs. Non-Hyperlipidemia)	−1.69 (2.65)
Kidney Stones	
Kidney Stones (vs. Non-Kidney Stone)s	−1.32 (2.34)
Smoking	
Smoking (vs. Non-smoking)	−1.24 (1.44)

* *p* < 0.05, ** *p* < 0.0001.

**Table 4 ijerph-19-04404-t004:** Summary of multiple regressions of for effect of plasma arsenic (As) and SNPs of TNF-α (different SNPs in models 1 to 9) on kidney function (CKDEPI eGFR), with all adjusted for BMI, diabetes, hypertension, hyperlipidemia, kidney stones, and smoking habits.

Dependent Variable	CKDEPI eGFR	β (SE)
Model 1	ln(Plasma-As)	−8.20 (1.71) **
	rs1799964	
	CC (vs. TT)	−0.05 (3.09)
	CT (vs. TT)	1.79 (1.38)
Model 2	ln(Plasma-As)	−7.98 (1.70) **
	rs1800629	
	AA (vs. GG)	−3.38 (3.80)
	AG (vs. GG)	−1.22 (1.61)
Model 3	ln(Plasma-As)	−7.93 (1.70) **
	rs1800610	
	AA (vs. GG)	2.86 (3.81)
	AG (vs. GG)	−2.42 (1.36)
Model 4	ln(Plasma-As)	−7.84 (1.70) **
	rs3093662	
	GG + GA (vs. AA)	0.65 (2.84)
Model 5	ln(Plasma-As)	−7.93 (1.70) **
	rs3093668	
	CC + CG (vs. GG)	−2.33 (3.43)
Model 6	ln(Plasma-As)	−7.74 (1.69) **
	rs769177	
	TT + TC (vs.CC)	4.02 (1.81) *
Model 7	ln(Plasma-As)	−7.83 (1.70) **
	rs769176	
	TC (vs. CC)	−2.71 (3.53)
Interaction models		
Model 8	ln(Plasma-As)	−6.47 (1.87) *
	rs1800629	
	AA (vs. GG)	−12.08 (20.67)
	AG (vs. GG)	11.41 (6.23)
	ln(Plasma-As) × rs1800629	
	AA (vs. GG)	6.93 (15.99)
	AG (vs. GG)	−9.59 (4.56) *
Model 9	ln(Plasma-As)	−8.29 (1.71) **
	rs769176	
	TC (vs. CC)	−41.32 (18.59) *
	ln(Plasma-As) x rs769176	
	TC (vs. CC)	28.83 (13.63) *

* *p* < 0.05, ** *p* < 0.0001.

## Data Availability

The data from Taiwan Biobank in this study can be accessed from the Taiwan Biobank at https://www.twbiobank.org.tw/new_web_en/about-export.php (accessed on 27 January 2022).

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
