# Peer review of "Interaction between Single Nucleotide Polymorphisms (SNP) of Tumor Necrosis Factor-Alpha (TNF-α) Gene and Plasma Arsenic and the Effect on Estimated Glomerular Filtration Rate (eGFR)"

_ijerph, 2022, doi:10.3390/ijerph19074404_

Round 1
Reviewer 1 Report
Some researchers have revealed that chronic As exposure would lead to positive regulation of TNF-α mRNA and other related factors [Balakumar and Kaur, 2009; Peters et al., 2015]. It means that there might be regulatory mechanisms in this case at transcriptome level. However, in the present study the authors have screened the SNPs at genomic level. Do you think if there is any association in between? I believe that still we need further study and further information to confirm the results. Please clarify.
The presentation of the results needs improvement. Conclusion needs improvement and should rewrite in a proper format. For example:
- The sentence “Our study that Plasma As decreased CKDEPI eGFR, which rs1800629 itself has the 271 property of regulating TNF-α, and had interaction with plasma As…” needs revision. Etc.
- The quality and the resolution of Figure 1b must be improved.
2- English language needs improvement. I suggest editing the MS by a native English Language service.
Reviewer 2 Report
The design of the study is quite poor. Authors should add more data to prove the finding of their study and conclusion.
Reviewer 3 Report
This study is interesting, but English needs more improvements.
Line 52. Minimal lethal dose or IC50 of As should be introduced around here.
Line 94." 。"should be " ."
Line 140. What are the differences among models 1-9?
Line 178. What was the standard was used for ICP-MS? and a reference is needed here.
Line 179. a reference is needed.
Line 193. Data analysis needs references.
Line 235. "。"
Line 238. "And rs769176 has not yet related research" reads confusing.
Line 263. voluntary bias needs a citation.
Line 271. 'Our study that' is a wrong sentence.
Line 273. No interaction was found.
Line 280. are
Model 1-9 should be clarified in the method section.
Round 2
Reviewer 2 Report
The manuscript is looking good and it can be accepted in the present format.
This manuscript is a resubmission of an earlier submission. The following is a list of the peer review reports and author responses from that submission.